# Electrotactile Feedback for the Discrimination of Different Surface Textures Using a Microphone

**DOI:** 10.3390/s21103384

**Published:** 2021-05-12

**Authors:** Pamela Svensson, Christian Antfolk, Anders Björkman, Nebojša Malešević

**Affiliations:** 1Department of Biomedical Engineering, Faculty of Engineering, Lund University, 223 63 Lund, Sweden; christian.antfolk@bme.lth.se (C.A.); nebojsa.malesevic@bme.lth.se (N.M.); 2Department of Hand Surgery, Clinical Sciences, Sahlgrenska Academy, University of Gothenburg and Sahlgrenska University Hospital, 405 30 Gothenburg, Sweden; anders.bjorkman@med.lu.se

**Keywords:** electrotactile feedback, texture sensor, non-invasive stimulation, friction sound, feature extraction

## Abstract

Most commercial prosthetic hands lack closed-loop feedback, thus, a lot of research has been focusing on implementing sensory feedback systems to provide the user with sensory information during activities of daily living. This study evaluates the possibilities of using a microphone and electrotactile feedback to identify different textures. A condenser microphone was used as a sensor to detect the friction sound generated from the contact between different textures and the microphone. The generated signal was processed to provide a characteristic electrical stimulation presented to the participants. The main goal of the processing was to derive a continuous and intuitive transfer function between the microphone signal and stimulation frequency. Twelve able-bodied volunteers participated in the study, in which they were asked to identify the stroked texture (among four used in this study: Felt, sponge, silicone rubber, and string mesh) using only electrotactile feedback. The experiments were done in three phases: (1) Training, (2) with-feedback, (3) without-feedback. Each texture was stroked 20 times each during all three phases. The results show that the participants were able to differentiate between different textures, with a median accuracy of 85%, by using only electrotactile feedback with the stimulation frequency being the only variable parameter.

## 1. Introduction

Every day the human hand is used to explore and interact with the surroundings. This is made possible by the delicate interaction between the sensory and motor systems in the peripheral and central nervous system. The human hand consists of 17,000 mechanoreceptors such as Meissner’s corpuscles, Merkel disks, Ruffini organs, and Pacinian corpuscles, located at different depths in the skin and they react to different stimuli [1]. They are categorized depending on the size of their receptive fields, adaptation rate, and location in the dermis. Exploring a texture with the fingers elicits texture-specific vibrations in the skin, activating both Pacinian and Meissner’s corpuscles which respond to high-frequency respectively to low-frequency vibrations [2].

Both the spatial pattern of the object manipulated and the temporal pattern with which the object is being manipulated play a role in texture perception. The different patterns are conveyed in afferent responses. The spatial, such as gratings and Braille dots (on the order of millimeters), evoke a response of slowly adapting type I (SAI) afferents. However, in discriminating natural textures the temporal pattern is more dominant. The temporal pattern is encoded in the responses of rapidly adapting (RA) afferents and Pacinian corpuscles [3].

The loss of a hand, through amputation, disconnects the afferent and efferent pathways from reaching their targets. In order to achieve the motor control necessary for object manipulation and object identification, the afferent pathways provide crucial information to close the loop between the hand and the brain and provide sensory feedback [4]. In addition, sensory feedback is essential for information about an object’s physical properties, such as texture, softness/hardness, etc.

Several different commercial prosthetic hands restore some motor functions to the amputee. However, these prostheses do not provide any sensory feedback and sensory feedback has been highlighted, by prosthesis users, as one of the desired functions in prosthetic hands [5]. It has been suggested that sensory feedback in a hand prosthesis should be modality-matched, meaning that pressure on a finger should be experienced as pressure by the amputee. Furthermore, the feedback should be somatotopically matched, meaning that pressure applied on, for example, the prosthetic index finger should be experienced as sensory stimulation on the index finger by the amputee. Somatotopically matched and modality matched sensory feedback mimics normal physiology and thus may reduce the cognitive burden that sensory substitution imposes on the prosthetic user [6]. Furthermore, it has been shown that adding sensory feedback facilitates the control of the prosthesis [7].

One way to provide sensory feedback is to use transcutaneous electrical nerve stimulation (TENS), a technique that is based on high-voltage electrical pulses sent through a pair or a plurality of electrodes placed on the skin, to stimulate nerve fibers. It is commonly used to relieve pain [8] or provide electrotactile feedback [9]. In addition, TENS can play an important role in the control of manipulation tasks for prosthesis users [10,11,12,13,14] and assisting in the interpretation of objects [15]. TENS applied to the skin over the median or ulnar nerve in the amputation stump result in sensations experienced as originating from the median or ulnar nerve innervated fingers in the lost hand (somatotopic sensation) [16]. Using TENS could aid prostheses users to discriminate a surface’s texture in a more intuitive manner and without sensory substitution, which is dominant in the case of other types of sensory feedback. Additionally, electrotactile feedback could potentially reduce phantom limb pain and stump pain in transtibial amputees [17] and also enhance the feeling of embodiment [18].

Technical solutions to provide sensory feedback of the force generated during a grasp are well explored, while feedback for texture perception for use in prosthetic hands is not. Interestingly, sensors used to detect texture-information are more common in self-organizing robots or robotics in applications such as health, eldercare, and manufacturing [19,20,21,22,23]. To provide an amputee with natural sensory feedback, implants that directly stimulate a peripheral nerve have been proposed. By using an artificial fingertip with a Micro Electro Mechanical System (MEMS) sensors using four transducing piezoresistors, the user could discriminate different textures based on the produced patterns of electrical pulses, which in turn, stimulate the nerves in the arm [24]. The small size and low power consumption in MEMS sensors are advantageous if used in a prosthesis. A proposed system for sensory substitution, to be used in prosthetic hands, used a force-torque sensor to obtain texture data from three different types of textures. By using a convolution neural network (CNN) algorithm, the different textures were classified and converted to vibrational stimulations [25]. Sensors based on Polyvinylidene Fluoride (PVDFs) films have been used for texture detection [26]. PVDFs, when stimulated by vibrations, display similar characteristics to fast adapting mechanoreceptors [22]. Yi et al. [27] developed a bioinspired tactile sensor based on piezoelectric materials, which was shown to mimic Meissner’s corpuscles. In addition, multi-modal sensors have been used to identify different materials, by implementing multiple gauge sensors, to capture resistance changes, together with PVDFs to capture electric potential changes. As mentioned, PVDFs are equivalent to fast adapting mechanoreceptors while the gauge sensors represent the slow adapting mechanoreceptors that detect lateral stretch, hence, it detects the static properties of a stimulus [22,23].

It has been shown that participants who have lost sensibility in a hand can substitute it, to some extent, with auditory information [28]. The participants could differentiate between different textures by listening to the friction sound picked up by small microphones. Another study used a microphone attached to the forearm to show that vibrations occurring, when the fingertip was sliding over a rough surface, can propagate from the fingers to the forearm [29]. This suggests that a microphone is a good candidate to pick up a texture’s acoustic characteristics. A classification analysis showed that the frequency composition in the texture-elicited vibrations consists of enough information to allow for the identification of different textures with high accuracy [30]. An early approach to texture recognition, a sensing pen was developed containing an electric microphone to classify different textures using neural networks [31]. A study showed promising results to use a capacitor microphone with an attached metal edge for texture sensing. When exploring different textures, the metal edge vibrates in different frequencies depending on the textural properties of the stroked material. The different textures could then be identified by using signal processing with the fast Fourier transformation (FFT), coupled with a supervised Learning Vector Quantization (LVQ) [32]. Another study implemented a node network of 10 microphones in robotic skin and classified different textures with a logistic regression model [33].

The current paper contributes a simple electrotactile feedback system with a computational method to convert recorded friction sounds, arising from stroking different textures, into somatotopic electrical stimulation in real-time. Electrical stimulation was chosen because of its easy application and the control of the stimulation parameters, such as the amplitude and frequency. The friction sound of a texture was recorded with a condenser microphone and median frequency was calculated of the audio spectrum. By analyzing the accuracy for discriminating different textures with the proposed system, conclusions can be made if the system is fit to be used as a texture sensing substitute. With the proposed system the participants had an overall median accuracy of 85% in discriminating different textures.

## 2. Materials and Methods

### 2.1. Participants and Ethics Approval

Twelve able-bodied participants, 10 males and 2 females (median age, 31 years; range, 24–44), with no known neurological disorders participated in the study. Two participants had previous experience of electrical stimulation, however, they were not familiar with the current study protocol. The rest of the participants had llittle knowledge of electrical stimulation in general. All participants provided informed consent, and the study was approved by the Swedish Ethical Review Authority (DNR 2020-03937).

### 2.2. Equipment

Four different textures [34] (see Figure 1) were used to evaluate the ability to discriminate between different surfaces based on electrotactile feedback.

An omnidirectional electret condenser microphone with an amplifier module (Adafruit MAX9814, Adafruit Industries, New York, NY, USA) (Figure 1a) was used as a sensor for the exploration of different textures, by picking up the friction sounds during stroking. The operating frequency range of the microphone was 20–20,000 Hz and the gain was set to 60 dBA. During the experiments, the experimenter was holding the microphone by hand while stroking the textures so that the enclosure was in direct contact with the surfaces (Figure 1b), thus, it was able to pick up the friction sound. The digitalization of the audio signal was done by a PJRC Teensy 4.0 microcontroller (32 bit 600 MHz ARM Cortex-M7 processor, using an NXP iMXRT1062 chip, PJRC.com, LLC, Sherwood, OR, USA). As the initial tests of the microphone-textures interaction showed that the friction-originated audio signal for the different textures was below 3 kHz, the microphone signal was sampled at 6 kHz with 16-bit resolution. The processing of the microphone signal and extraction of the signal features that depicted characteristic vibrational/audio responses during the tactile exploration was done in real-time by Teensy.

The extracted signal features were sent to a custom-made electrical stimulator capable of producing biphasic charge-balanced cathodic-first current-controlled pulses of amplitudes in the range from 0.1 mA to 10 mA (steps of 0.1 mA), and frequencies of 1 to 100 Hz. The DC/DC boost switching regulator was used to generate stimulation voltage which was maximally 38 V (depending on the skin impedance and the stimulation current). The stimulation control was done by an onboard PIC18F25K22 microcontroller (Microchip Technology, Chandler, AZ, USA). The microcontroller managed generation of the stimulation patterns and communicated with both, the Teensy microcontroller and PC, enabling alteration of stimulation parameters in real-time. The electrical stimulation was delivered to the participants through self-adhesive Pals electrodes (Axelgaard Manufacturing Co., Lystrup, Denmark), placed on the skin over the median nerve so the sensations following stimulation, were associated with the median nerve innervated fingers (thumb, index, and the middle finger) and palm area. An overview of the system can be seen in Figure 2. The PC included in the setup had a non-essential role as it was used just to initiate the protocol and visualize stimulation frequency in real-time during the experiments.

### 2.3. Algorithm

The formulation of the algorithm presented in this paper was based on several empirical pre-tests (the algorithm evaluation stage) that were used to characterize friction-based interaction between the microphone and selected textures. These pre-tests identified the audio frequency range resulting from friction with selected textures and the behavior of several signal features, such as total signal power, peak frequency, mean frequency, and median frequency, during manual stroking.

The first step of the algorithm that was designed and implemented, as a part of this study, was the calculation of the frequency content of the microphone signal. The FFT calculation was performed on 2048 samples of the digitalized audio signal using a modified version of the Arduino library (http://github.com/kosme/arduinoFFT, accessed on 11 November 2020). The FFT was updated after every 128 samples, corresponding to a ∼50 Hz update rate with the sampling rate of 6 kHz. In the next step, the spectral components at 50 Hz and below 20 Hz were removed from the FFT spectrum.

The feature of the microphone signal that was heuristically chosen, was the median frequency of the audio spectrum. Besides this feature, several other well-known features, such as the mean frequency and the audio signal envelope, were tested in a small sample trial, but the median frequency showed the best results in discriminating different textures. In the next step, the median frequency was linearly translated into stimulation frequency. The rationale for devising the transfer function was to shift the median frequencies of the audio signal to the range of stimulation frequencies that are commonly used for sensory/neural stimulation [35,36]. The devised transfer function equation was:(1)stim=(medianf−lowerB)/scaling+5,
where *stim* denotes the stimulation frequency (in Hz) that was sent to the stimulator, *medianf* denotes the median frequency signal feature, *lowerB* denotes the lowest median frequency that was empirically chosen as relevant, and *scaling* denotes the linear scaling factor used to constrain the possible median frequencies of the microphone signal into the range of stimulation frequencies produced by the stimulator. In the current study, the *lowerB* and *scaling* constants were empirically set to 50 and 10, respectively. The addition of 5 Hz was done to constrain the lower dynamics of the stimulation as extremely low frequencies would significantly reduce the information bandwidth delivered to the participant. In other words, sending a low stimulus frequency (e.g., below 1 Hz) would mean that the next stimulation pulse, and also the change of stimulation frequency, would have to wait for a long time (more than 1 s in the case of <1 Hz stimulation). This is the result of the electronic stimulator protocol which accepts updates only after producing a stimulation pulse specified by the last command. In addition, the stimulation frequency resulting from Equation (Equation 1) was constrained to 80 Hz, thus frequencies calculated as higher than 80 Hz were set to 80 Hz. The total processing time of the system is calculated to be 220 ms, which includes the communication delay (UART) of 20.6 ms, and due to the waiting until the last desired pulse is generated (max 200 ms in the case of 5 Hz stimulation). The processing time is considered to be fast enough to be considered “real-time” and should not affect the results as the reaction time to sensory stimuli is 50–300 ms [37].

Apart from the calculation of the stimulation frequency, the algorithm extracted the total signal energy from the microphone signal over the last 2048 samples, by summing the spectral components from 1 to sampling_frequency/2, which was used to enable/disable the stimulation. The stimulation was activated in the case of the magnitude exceeding the empirically chosen threshold, in this case, 300 mV. The threshold was set so that the stimulation was active only while the microphone was in contact with a surface.

The medians of the audio signal can be seen in Figure 3, with an error which is set to lower and upper quartiles of 25% and 75% respectively. As seen in the figure, there is some overlap between median frequency patterns resulting from stroking different textures. The silicone has a median frequency which gradually ramps up in frequency during the stroke, sponge and felt had a median frequency of approximately 200 Hz and 50 Hz, respectively while the mesh has a varying median frequency. The frequency responses presented in Figure 3 were obtained during the evaluation stages of the algorithm when the experimenter stroked the textures in an unobstructed (without any disturbances between stroking) and paced manner (the visual cue for executing strokes was given every 10 s). In the experiments with participants, the experimenter had to wait between consecutive strokes until the participant responded, which reduced repeatability leading to inconsistent stroke duration and dynamics. Another important difference between the algorithm evaluation stage and the study was that the median frequency update interval in the case of the evaluation stage was constant (∼50 Hz) while during the experiment the update was constrained by the latest stimulation frequency, thus ranging from 50 Hz to 5 Hz. This limitation was particularly noticeable when the median frequency of the audio signal was low, e.g., while the sponge or mesh was being stroked, and during the initial part of the stroke in the case of silicone rubber.

### 2.4. Protocol

The experiment consisted of three phases: (1) Training phase, (2) feedback phase, and (3) without feedback phase. All phases were performed with participants sitting down in a quiet room. The initial step of the protocol was the placement of the stimulation electrodes. The anode (rectangular 7 × 10 cm Pals electrode) was placed on the ventral side of the forearm, while the cathode (round 2.5 cm in diameter Pals electrode) was placed over the median nerve, just proximal to the wrist as shown in Figure 1c. In this area, the median nerve is located between the tendon of the flexor carpi radialis longus muscle and the tendon of the palmaris longus muscle. Next, a stimulation at 50 Hz, with a duration of 2 s was applied to identify a sensory threshold, pain threshold, and the presence of the somatotopic sensation due to the nerve stimulation. After finding the sensory threshold with the resolution of 0.1 mA, the current intensity was increased in steps of 1 mA until reaching a painful level of stimulation. As the current output of the electronic stimulator was limited to 10 mA, the pain threshold was also capped at 10 mA. The amplitude of the current pulses that were used in the experiment was set to level = 3 (on a scale between 0 = no-sensation and 10 = maximum pain level), where the sensory threshold is level = 1 and the pain threshold level = 8. Level = 3 is considered to provide a distinct stimulation without evoking any pain. The stimulation amplitude for each participant can be seen in Table 1. At the current amplitude selected for the experiment, each participant was asked to inform the examiner about where he/she perceived the stimulation. If the stimulation was perceived in median nerve innervated skin areas, such as the thumb, index finger, middle finger, and part of the ring finger, but not at the electrode location, the stimulation was considered somatotopic. In the case of non-somatotopic sensation, the cathode was slightly relocated, and the previous steps were repeated until an electrode location leading to somatotopic sensations in median nerve innervated fingers were found.

Upon establishing the stimulation amplitude, the training phase of the experiment was initiated. In this phase, the experimenter sequentially stroked each texture for 20 cycles (in total 80 strokes) with an estimated speed of 14–25 mm/s. During this phase, the participant got to watch the strokes and at the same time receive the electrotactile feedback.

After finishing 20 cycles, the participant was blindfolded and acoustically insulated using headphones, and the stimulation with the feedback phase was initiated. During this phase, the experimenter stroked the different textures in a proximal to the distal direction in a randomly predefined sequence, while the participant was instructed to verbally identify the texture. Up to two additional repetitions of the same texture were allowed, if requested by the participant. Upon receiving a response from the participant, the experimenter provided verbal feedback consisting of true/false statements and the information regarding the stroked texture (in the case of false response by the participant). This phase consisted of 20 repetitions of each texture (80 in total). Upon completing the phase, a short break was taken (approximately 5 min).

The final phase also comprised 20 repetitions of each texture in a new randomized order, but without feedback from the experimenter. As in the previous phase, two additional repetitions of the same texture were allowed for the participant.

It should be noted that all of the strokes were subjected to variability due to the manual execution by the experimenter. Specifically, with sticky or rough surfaces, such as silicone rubber, the vibrations resulting from the friction were unpredictable. In the case of smoother surfaces, the speed and consistency of the strokes were also directly mirrored in the frequency of the stimulation.

### 2.5. Data and Analysis

Two separation analysis was done in this study, one to analyze the consistency data and one on the following experimental data.

A custom-made LabVIEW program (Labview 2018, National Instruments, Austin, TX, USA) was used to record data during the evaluation stages of the algorithm development. These tests comprised stroking of each texture pseudorandomly by the experimenter and were used for further investigation of the consistency of the stroking. Two factors were explored; the consistency of the time of each stroke and the consistency of the frequency. The data analysis was performed in Python with several libraries, such as Scikit-learn (https://scikit-learn.org/, accessed on 11 November 2020) and SciPy (https://www.scipy.org/, accessed on 11 November 2020).

A Generalized Linear Mixed Effects model was fitted to the the data using jamovi [38,39,40]. The dependent variable was the accuracy of the responses of the subjects per phase and rod. A Poisson distribution using a log link function was used as this fits the type of data. Phase (with feedback and without feedback) and stimulus type (mesh, felt, sponge, and silicon rubber) were the factors and levels of the experiment. The participant was considered a random effect. Subsequent post-hoc comparisons for the different textures within each phase using Z-tests were corrected using Holm’s sequential Bonferroni procedure.

## 3. Results

The accuracy of all 12 participants in identifying four different textures are shown in Figure 4a. It should be noted from this figure that the variance of accuracy for different participants is relatively large. Three of the participants had an accuracy higher than 90% in the experimental phase with verbal feedback, while this number increased to six participants during the last phase when no feedback was provided to participants. As the accuracy is not normally distributed, the total accuracy was calculated as the median of individual accuracy. The total median accuracy for all participants and textures was 85% (IQR 70–95%). The participants needed an average of 1.54, 1.24, 1.44, and 1.40 repetitions for each texture (silicone rubber, felt, sponge, and mesh) before responding to the stimulus generated by the different textures. The overall performance for each texture can be seen in Figure 4b. There was a statistically significant difference in terms of phase (with feedback, without feedback, *p* = 0.034) and texture (*p* < 0.001). The multiple comparisons test performed on textures per phase revealed there was a statistically significant difference between some of the textures, namely felt vs. silicone rubber in the with feedback condition (*p* < 0.001), mesh vs. silicone rubber in both feedback conditions (*p* = 0.016 (w/ FB), *p* = 0.0086 (w/o FB)), and sponge vs. silicone rubber in the no feedback condition (*p* = 0.043). For the other combinations there were no statistically significant differences.

The experimental results can be seen in the stimuli-response confusion matrix in Figure 5. During the final phase (without feedback) the median performance for each texture: Silicone rubber, felt, sponge, and the mesh was 77.5% (IQR 62.50–96.25), 90% (IQR 68.75–100.00), 97.5% (IQR 73.75–100.00), and 92.5% (IQR 85.00–100.00), respectively.

During the second phase (stimulation with feedback), felt was often mentioned, by the participants, as being the easiest to distinguish (85.4%) because of its higher median frequency. However, it was sometimes confused with silicone rubber (12.9%), since silicone rubber, on some occasions, gave inconsistent stimulation because of its sticky characteristics in the texture. This made the participants, on some occasions, hesitate if the stimulation stemmed from silicone rubber. Vice versa, silicone rubber was in 15% of strokes misidentified as felt. As mentioned, silicone rubber gave a low frequency at the beginning, which increases during the stroke, however, during manual stroking on few occasions the audio signal did not have this characteristic and instead was confused with the felt 15% of the time and 12.5% of the time it was misinterpreted as the sponge. In the final phase (stimulation without feedback), the performance improved for mesh, sponge, and silicone (+7.1%, +6.7%, and +5.9% respectively) while the ability to detect felt decreased slightly (−4.2%).

## 4. Discussion

The presented study was designed to assess the feasibility of developing a computational method for the direct conversion of the sound detected by a microphone, when stroking a texture, into an electrotactile stimulation pattern that could be used to distinguish between different textures. However, the microphone and the audio amplifier used in this study were regular, off-the-shelf components, not chosen for the specific purpose of measuring contact vibrations. It should also be noted that the experimental conditions in this study were designed to resemble conditions that would be expected in a real-world application of a feedback system in a hand prosthesis. Mainly, the sound was translated into the frequency of the electrical stimulation continuously, permitting variability of the feedback, in accordance with the natural variability in stroking velocity and pressure seen in a hand exploring an object.

The results of the study are encouraging since, after just a brief familiarization period comprising 20 strokes of each texture which lasted approximately 5 min in total, participants were able to achieve a relatively high overall median accuracy (85%). It should be pointed out that only two participants had any previous experience with electrical nerve stimulation which makes the familiarization of only 5 min even more rigorous in the case of naive participants. These participants had to get accustomed to both, the new sensation in general and the variation of stimulation frequency due to the interaction with different textures. It should be emphasized that the experiment was designed such that the participants had their vision and hearing occluded while in a real-life exploration of textures, the vision and tactile perception plays an equal role in identifying textures [41]. The same would apply to the auditory cues. When exploring a texture the sound of the stroking of textures can also help to perceive a texture’s roughness [42]. It has also been shown that auditory cues could be more beneficial than visual and tactile cues to detect a material’s stickiness [43].

Jamali and Sammut [22] used PVDFs to detect vibrations from materials for the classification of seven different surface textures based on three and five Fourier coefficients and with 50 learning samples per each. The stroking was done by a robot ensuring high repeatability of strokes. Furthermore, the results of Jamali et al. are intended for machine-based classification only, without involving human participants as recipients of the feedback information. The prediction accuracy for their algorithm, using a naive Bayes learner, was 78% when three Fourier coefficients were used and 83.5% when five coefficients were used. Compared to our study, the median accuracy of a human of 85% is a promising result for providing continuous feedback to participants. The same group [23] presented another method based on the learned classifier, resulting in a higher accuracy of 95% ± 4% on the unseen data. The setup consisted of an artificial robotic finger with implemented sensors that respond to stretch (strain gauges) and vibration (piezoelectric sensors). However, having a robotic finger with set pressure and velocity, the stroking of the material will be highly consistent, thus it could be debated if a learned classifier used in a controlled laboratory environment would perform as well in real-life manual stroking. Hughes and Corell [33] did consider the inconsistency in a human operator in their study, by stroking the textures by hand to include the variability in speed and pressure of the stroking. They implemented a network of sensor nodes, using omnidirectional microphones, embedded in silicone rubber for texture recognition showing that the skin prototype was able to identify 15 different textures with an accuracy of 71.7%. It should be noted that all of the aforementioned studies referenced within this paragraph present results on the ability of machine learning algorithms to discriminate between textures. It has not been shown if a human participant could match such performance in a real-time feedback setup (as presented in our study).

The present study could also be considered as the worst-case scenario as the feedback is directly proportional to the texture stroking dynamics which was completely governed by the experimenter, while the participant did not have any complementary information. Thus, we hypothesize that implementation of the concept of the proposed system, using a microphone as a sensor and electrical stimulation as a feedback mechanism, to provide information about a texture would significantly improve the accuracy of identifying a texture. The results support this hypothesis, where the accuracy is significantly improved between the second phase (with feedback) and final phase (without feedback) which was done in a short-term controlled experiment. Considering that the system will be used long-term and during activities of daily living, the user then would also use their natural feedback modalities, such as audio, visual, proprioceptive, or force, at his/her disposal. The user would then be able to incorporate his/her natural feedback modalities to further strengthen the internal models of interaction with different textures. Furthermore, with the continuous use of the proposed electrical feedback system, texture recognition would likely gradually improve because in our data there is a significant improvement in performance even between two consecutive tests. Hence, it could be argued that during long-term passive learning, the overall accuracy could be improved for the proposed system [44]. In addition, prolonged use accompanied with the spacing effect, where learning is spread over time [45], could also enhance overall accuracy.

One of the major sources of errors identified by the experimenters and participants was the inconsistency of the stimulation pattern dynamics for some of the textures. Manual stroking is prone to variations in applied force, stroking velocity, and velocity profile, all of which affect friction sound, and consequently electrical stimulation. However, all of these parameters are also subject to variation in the normal use of hands. The accuracy during the tests (which were between 70% and 89%) are the consequence of the high variability in median frequency of the audio signal resulting in the partial overlap between friction responses of other textures (see Figure 3a). In addition, the acoustic signals for the different materials might also deviate for each participant depending on where on the fabric/textures the strokes were applied. These deviations in signal were mainly noticeable for the non-smooth textures, such as mesh and silicone rubber. This can be seen in the results, showing a statistical significance in accuracy between silicone rubber and the other smoother textures (felt (w/ FB) and sponge (w/o FB)) supporting the discussion that a possible explanation for lower accuracy in discriminating silicone rubber due to its stroking inconsistency. There was also a significant difference in accuracy between silicone rubber and mesh indicating that despite the non-smooth texture mesh was easier discriminating compared to silicone rubber, whilst there were no significant difference between mesh and the other textures (felt and sponge), hence it can not be concluded that smoothness is crucial in being able to discriminate the textures. In the case of mesh texture, the distance between knots of the fabric was not constant, but upon rhythmic, consistent stroking this texture could resemble the sponge which was characterized by a low median frequency. In the case of silicone rubber, the sticky texture often resulted in the median frequency ramping-up (as shown in Figure 3a), but due to inconsistencies in applied force, this characteristic signal feature was sometimes missing, making it difficult to distinguish silicone rubber from felt or sponge. Both of these issues are the consequence of separation/partitioning between movement and sensation (the movement was performed by one person and the sensation is experienced by another person). This would be minimized in a real-world scenario where the same person does the movement and receives the sensory feedback, thus employing an internal forward model of hand movement and experiencing movement dynamics with complementary senses, such as proprioception, force, and vibration feedback (natural or externally generated). It can also be noted that the stimulation frequency increases with the stroking speed, and this applies also to the human hand where Manfredi et al. [30] recorded the skin vibrations during exploration of textures. The recordings showed an increase in frequency with increasing speed, and this applies both to non-periodic and periodic textures. Thus, having the same person performing texture exploring and perceiving the sensation, proprioceptive information will aid the central nervous system (CSN) to determine the velocity of the moving limb [46], in this case, the information of the stroking speed. Hence, the felt stimulation frequency could be associated with the stroking speed.

Due to the omnidirectional feature of the microphone, it picks up sound with equal gain in all directions, making it susceptible to background noises which is a limitation associated with the study. However, the experiment was performed in a quiet room which eliminated most of the background noises. If the currently described system was to be used for a prosthetic hand, background noise could be removed by using a noise-canceling sub-system where an additional microphone could detect only background noise.

Having in mind that other feedback modalities, such as force or hand aperture, would be prioritized in a prosthetic hand as they are coupled with the basic prosthetic hand functionality (grasping), the goal of this study was to evaluate the feasibility of using appropriate electrical stimulation for texture discrimination feedback using a microphone to pick up friction sounds of textures. As the feedback related to different sensors, such as force sensors, encoders, and microphones, would be combined within the same feedback interface (same electrical stimulator and electrodes), it was decided to dedicate only a single controllable stimulation parameter (frequency) out of many, such as, stimulation amplitude, frequency, pattern, and location, to texture exploration. Thus, frequency modulation of the electrical stimulation delivered to one cathode was chosen as a minimalistic setup, leaving other parameters available for other potential feedback modalities.

The presented system is designed to be portable. The hardware components comprising sensor and actuator sub-systems are based on two microcontrollers, ARM Cortex M7 and PIC18F25K22. The former processor handles audio signal sampling and execution of the algorithms presented in the paper, while the later processor is responsible only for executing electrical stimulation in a time-crucial manner. Additional circuitry related to the stimulator analog output stage, consisting of the step-up converter and discrete components, has a relatively small footprint (less than 4 cm^2^ in the current version) and power consumption. Therefore, this hardware setup could be implemented within common hand prostheses by (1) integrating one or several miniature microphones on the prosthesis fingertips within a silicone glove, (2) placing at least one pair of electrodes over one of the major hand nerves, and (3) embedding all necessary electronics (including the prosthesis control part) on a single printed circuit board. As the presented system is self-contained, it could be integrated with existing and future powered hand prostheses or even in cosmetic prostheses.

## 5. Conclusions

This study presented an electrotactile feedback system with a microphone as a sensor, making it possible to pick up friction sounds from textures. In addition, a simple computational method to convert the signal transduced by the microphone into electrical stimulation was developed. The median frequency was calculated on the transmitted signal, since this feature had the best results in discriminating textures. The system provided the participant with somatotopic electrical stimulation from the processed microphone signal, which resulted in the participants being able to identify differences in textures. To the best of our knowledge, this concept is novel and there are no similar studies with the proposed system reported in the literature. The goal of this research was to devise an algorithm and self-contained hardware capable of supplying continuous feedback during texture exploration, and with future improvements, it would be interesting to investigate the performance during long-term use by a prosthesis user. The presented paradigm offers a unique feedback modality as there are no constraints regarding the number of detectable textures or their properties while the particular stimulation patterns resulting from stroking different textures could be learned by a user over time. In addition, the learning curve was steep, illustrated by the accuracy of 85% in participants (who had no prior knowledge of electrical stimulation) identifying different textures already after 20 repetitions.

## Figures and Tables

**Figure 1 sensors-21-03384-f001:**
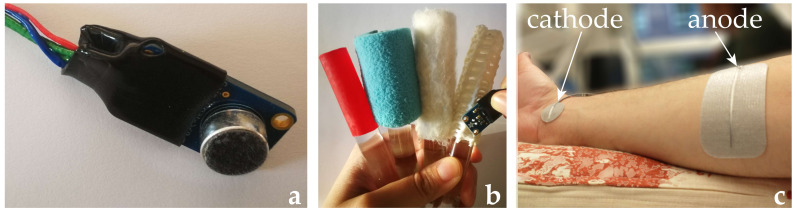
(**a**) Close-up image of the omnidirectional electret condenser microphone with an amplifier module. An isolation cable was put on the circuit board for an easier grip of the microphone during the stroking and to reduce interference with the components on the printed circuit board (PCB). (**b**) The experimenter was stroking the different textures with a microphone in a proximal to distal direction. (**c**) Placement of the electrodes on the participant’s forearm. The cathode was placed over the median nerve while the anode was applied on the upper part of the forearm.

**Figure 2 sensors-21-03384-f002:**
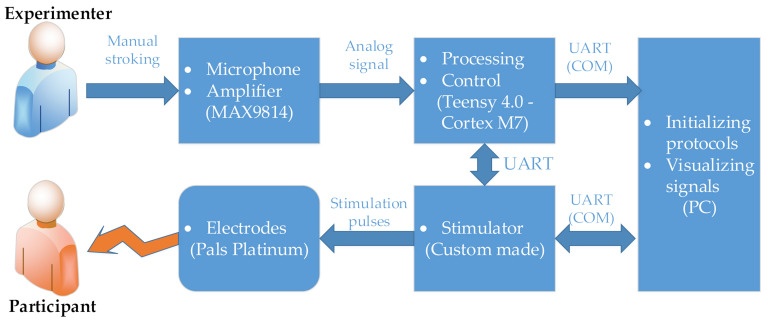
System overview. The manual stroking was performed by the experimenter with an omnidirectional electret microphone with the integrated amplifier. The audio signal was digitized by a Teensy 4.0 microcontroller for further signal processing and signal features extraction. The calculated median frequency was then sent to the custom-made electrical stimulator through a Universal Asynchronous Receiver/Transmitter (UART) connection. The electrical stimulation was then delivered through self-adhesive Pals electrodes attached to the participant’s forearm skin over the median nerve. Both, Teensy and the stimulator were communicating with a PC using their UART connections.

**Figure 3 sensors-21-03384-f003:**
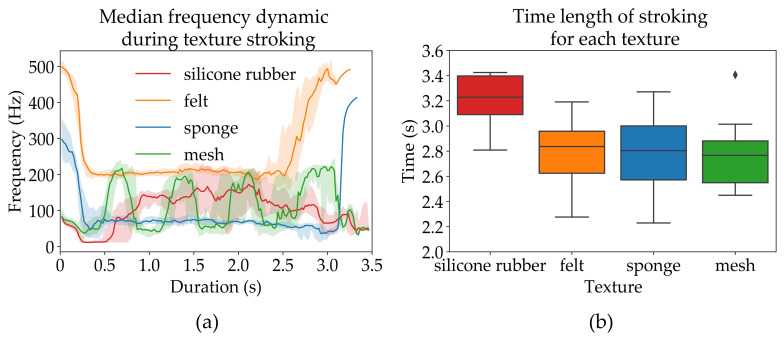
Consistency-test for the four textures. A total of 20 strokes were applied on each texture. (**a**) Median frequency error, set with lower and upper quartile (25% and 75%). There was a better consistency for felt and sponge, but the error is bigger for the silicone rubber and mesh. (**b**) The time consistency during the strokes.

**Figure 4 sensors-21-03384-f004:**
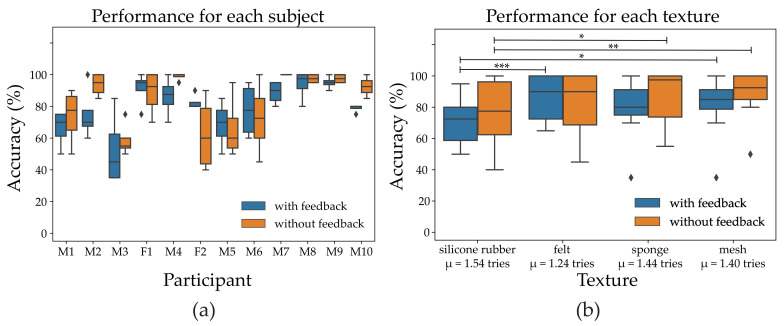
(**a**) The median accuracy for each participant where three participants were over 90% during the first phase (with feedback) and six participants during the second phase (without feedback). (**b**) The box plot shows the performance of the 12 participants on each texture. The x-axis also shows the average attempt the participants had for each texture.

**Figure 5 sensors-21-03384-f005:**
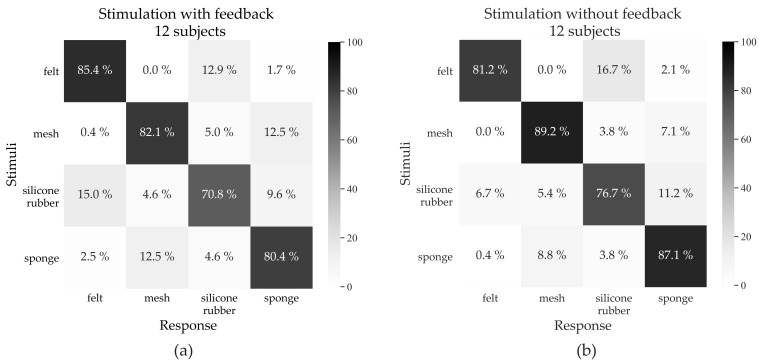
Confusion matrix for the identification of the different textures during the two phases. (**a**) Phase with feedback, where felt had the highest accuracy (85.4%). The lowest accuracy was when discriminating silicone rubber (70.8%), which was occasionally misinterpreted as felt (15.0%) and as sponge (9.6%). (**b**) Phase without feedback, where mesh had the highest accuracy (89.2%). In general the performance increased for all textures except felt in the phase without feedback. Mesh is easiest to discriminate (89.2%), sponge has also a high accuracy (87.1%). There was also a small improvement for the silicone texture (76.7% vs. 70.8%).

**Table 1 sensors-21-03384-t001:** Participants in the study and the individually-set levels of stimulation amplitude (mA) where the stimulation frequency was 50 Hz and with a pulse duration on 250 μs. The perception threshold (Level 1) is the just-noticeable stimulation amplitude. The pain threshold (Level 8) is when the amplitude is too high for the participant to endure. The stimulation amplitude chosen for this study (Level 3) is the amplitude that was considered to provide the participant with a distinct stimulation that was also comfortable for prolonged exposure.

Participants	Perception Threshold (Level 1)	Stimulation (Level 3)	Pain Threshold (Level 8)
M1	3.2	5.1	10.0
M2	2.2	3.9	8.0
M3	2.8	3.4	5.0
F1	3.0	5.0	10.0
M4	3.1	5.1	10.0
F2	2.1	3.8	8.0
M5	1.3	2.1	4.0
M6	2.3	4.2	9.0
M7	2.3	3.4	6.0
M8	1.7	3.8	9.0
M9	2.0	4.0	9.0
M10	3.5	4.8	8.0

## Data Availability

The data presented in this study are available on request from the corresponding author. The data are not publicly available due to ethical restrictions.

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
