# Peer review of "Electrotactile Feedback for the Discrimination of Different Surface Textures Using a Microphone"

_sensors, 2021, doi:10.3390/s21103384_

Round 1

Reviewer 1 Report

I found your study contained in the paper interesting, as a first step towards the technogical use of portable system for identification of some material when using a prosthetic hand.

In my opinion, your text is very readable and understandable, but I would like to have some clarifications or explanations on minor details:

  • In the text (line 56) you cited "citedupan2020temporal". Could you clarify it?
  • Concerning the protocol: you selected 12 participants. Is this number enough? which is the reason for selecting this number?
  • Concerning the experiment: why you determined the concrete number of cycles (20)? Is there any reason coming from other previous studies?
  • You mentioned a break of 5 minutes between the different phases of the experiment. Is this enough in order to prevent "memory"? It seems a short time in order to prevent and avoid biasing...

I suggest you to explain a little bit better the objective and contents of the section 2.5. May be a short introduction on the future use of the recorded data in relation to their analysis.

Reviewer 2 Report

Typo: P2 line 56 provide electrotactile feedback citedupan2020temporal.

The paper extensively describes the experimental set-up. However the discussion section is very long and very speculative, which in my view does not add much to the main message of the paper.

Reviewer 3 Report

The paper presents results from experiments that collect data from a microphone that picks up friction sounds from four different textures. The paper is well written, the technical content is right as well as the experimental procedure. It also provides enough tutorial content and discussion of the results in the context of previously reported works. The limitations and possibilities of the proposal and experimental procedure are also extensively discussed.

Some comments are:

  • I do not see clearly the interest in using a microphone as sensing device, since, as you point out in the manuscript, it can collect other sources of noise that can interfere. The acoustic signals can be collected by other sensors such as accelerometers or pressure sensors (for instance in the bio tact -https://syntouchinc.com/). Nevertheless, a clear advantage is the cost, as long as a common off the shelf device is used. It is more interesting to me the fact of developing the transfer function and the feedback and evaluate it.
  • Please argue why the variable time between consecutive strokes affects the repeatability (lines 202-204). It obviously affects the repeatability of the experiment, but I do not see clearly why it has to affect the repeatability of the results.
  • I think it could be better if the use of the median frequency as the basis of the transduction method is mentioned in the Conclusions section.
  • The processing time should be provided in the results section. I guess it could affect the performance of the procedure if the participant explores the surface by him or herself because it could introduce a noticeable delay.
  • Line 56 (page 2) – “citedupan2020temporal”
  • The quality of Figure 4(b) in my pdf is not good.
